

# ClimarisQ: What can we learn by playing a serious game for climate education?

Davide Faranda(1,2,3), Lucas Taligrot (1), Pascal Yiou (1), Nada Caud (1)

(1) Laboratoire des Sciences du Climat et de l'Environnement, UMR 8212 CEA-CNRS-UVSQ, Université Paris-Saclay & IPSL, CEA Saclay l'Orme des Merisiers, 91191 Gif-sur-Yvette, France

(2) London Mathematical Laboratory, 8 Margravine Gardens, London W6 8RH, UK

(3) LMD/IPSL, ENS, Université PSL, École Polytechnique, Institut Polytechnique de Paris, Sorbonne Université, CNRS, 75005, Paris, France

*Correspondence to: Davide Faranda (davide.faranda@lsce.ipsl.fr)*

***Abstract:*** Climate change education faces the twin challenges of conveying complex scientific concepts and inspiring urgent action. ClimarisQ is a web and smartphone-based serious game developed by the Institut Pierre-Simon Laplace (IPSL) to address these challenges by simulating climate–societal dynamics and extreme events in an interactive format. This article evaluates ClimarisQ's role as an innovative educational tool to raise awareness of climate issues. We outline the game's design (grounded in real climate models and IPCC scenarios) and its pedagogical objectives of illustrating the urgency of collective action, the complexity of climate-ocean interactions, and the ethics of decision-making under uncertainty. We present results from a user questionnaire (77 respondents) assessing learning outcomes and user feedback. Players rated the game highly in terms of usability, scientific content, and engagement (average 4.2/5 across categories), and qualitative feedback indicates that ClimarisQ effectively fosters discussion and systems thinking about climate challenges. However, many already knowledgeable players reported learning few new facts, highlighting the need to tailor content to varying prior knowledge. We discuss the strengths of ClimarisQ – notably its ability to simulate feedback and extreme events in an accessible way – and its challenges, such as balancing scientific accuracy with playability and ensuring inclusivity. Situating ClimarisQ in the broader context of climate outreach, we compare it with other educational games and initiatives. We emphasize the



ethical responsibility of climate communication tools to empower action without misinformation or
fatalism. In conclusion, ClimarisQ demonstrates how serious games can complement formal education
and engage diverse audiences in climate-ocean literacy, an approach that is increasingly vital given the
urgency of the climate crisis.
*Keywords*: climate change education, serious games, science communication, extreme events

## 1. Introduction

Climate change is a complex, "wicked" problem characterized by nonlinear interactions, long
timeframes, and global scope (Levin et al., 2012). The interactions between humans and the climate
system are inherently complex, involving nonlinear feedbacks, time lags, and cross-scale dynamics
(Steffen et al, 2007). Traditional information-based approaches have often struggled to translate
scientific knowledge into public engagement and action, as greater awareness does not automatically
lead to behavior change (Whitmarsh et al., 2021). Many people find it hard to connect with climate risks
that feel abstract or distant from daily life, especially if they have not personally experienced climate
impacts (Spence et al., 2011). In this context, educators and communicators are exploring creative,
interactive strategies to foster deeper understanding and commitment (Monroe et al., 2019). Serious
games – games designed for purposes beyond entertainment – have emerged as a promising tool to
bridge the gap between scientific complexity and public understanding of climate change (Flood et al.,
2018; Rooney-Varga et al., 2018). Over the last decade, there has been a growing interest and a
substantial increase in research on using serious games for sustainability and climate education
(Ahmadov et al., 2024). Serious games offer an engaging, experiential form of learning that can
complement formal climate education and science communication efforts. Unlike conventional didactic
methods, games immerse players in simulated environments where they can experiment with decisions,
face consequences, and learn by doing. Dozens of climate-related serious games have been developed,
ranging from board games and role-play simulations to computer and mobile games (Reckien &
Eisenack, 2013). These games target various aspects of climate change – from mitigation of greenhouse
gas emissions to adaptation to impacts – and audiences from students to policymakers. Reviews of the



field have found that climate change games have proliferated globally and cover multiple scales and
topics (Reckien & Eisenack, 2013; Flood et al., 2018). Crucially, studies report that well-designed
games can facilitate social learning, dialogue, and systems thinking around climate issues, often
succeeding where purely informational approaches fall short (Flood et al., 2018). Serious games create
a safe space for players to explore the complexity of climate systems and policies, make mistakes, and
observe outcomes, which can enhance understanding and motivation to act (Rooney-Varga et al., 2018).
For example, a systematic review of 46 studies concluded that the impact and value of serious games
for climate adaptation have been overwhelmingly positive, citing their ability to build trust among
stakeholders and generate enthusiasm for learning (Flood et al., 2018). Likewise, a recent review of
climate change education research highlights the need for participatory, affective, and innovative
approaches – exactly the strengths of serious gaming (Monroe et al., 2019).
ClimarisQ is a new serious game developed as a scientific outreach project to engage players with the
complexity of the climate system and the societal challenges of facing climate extremes. It joins a
lineage of climate games that aim to translate abstract scientific concepts into interactive non-
judgemental experiences. ClimarisQ is a web and mobile game in which players take on decision-
making roles to limit greenhouse gas emissions and manage resources in the face of increasing extreme
weather events. By design, the game incorporates authentic scientific data (from climate models) and
simulates feedback mechanisms between the climate, the economy, and society. The game's design
requires balancing economic, social, and environmental goals, thereby demonstrating trade-offs and
feedback loops that political leaders have to face (e.g. how investing in green infrastructure might
reduce future disaster losses but could short-term strain budgets or public approval). It also incorporates
the inherent uncertainty and variability of extreme events – these events occur with unpredictable timing
and location, though with frequencies influenced by the climate state. This feature teaches that while
individual events cannot be precisely predicted, statistical risks *do* increase with climate change,
echoing real-world climate science where attribution of extremes is probabilistic. ClimarisQ serves as
a tool to discuss the ethical dimensions of climate decisions. In the game, players' choices (such as
prioritizing economic growth vs. environmental protection) can prompt reflection on justice and



responsibility: who benefits or suffers from certain policies, and what ethical stance should guide
decision-making under uncertainty? Such discussions can be facilitated by teachers or moderators
during gameplay, connecting the simulation to real debates about climate justice, intergenerational
equity, and global responsibility – key themes in the special issue on practice, ethics, and urgency.

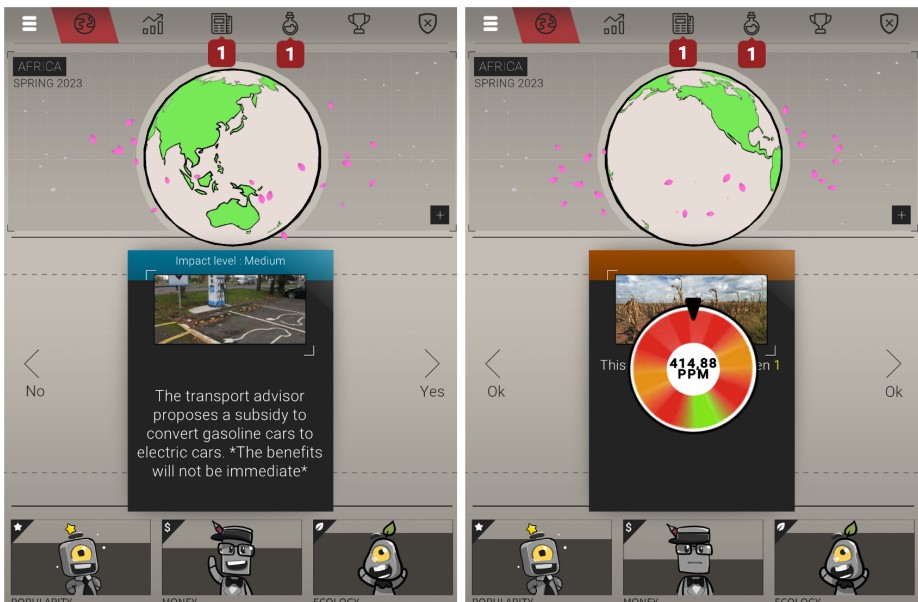


*Figure 1: Screenshots from ClimarisQ, showcasing interactive decision-making scenarios. The left*
*screen displays a transport advisor's proposal to switch from gasoline to electric cars. This decision*
*impacts three key parameters: Ecology, Popularity, and Money. Players must balance these elements,*
*as any parameter reaching zero will result in game over. The right screen shows the current CO2*
*concentration (414.88 PPM), which influences the number of extreme weather events in the game.*
*Higher CO2 levels increase the likelihood of such events, affecting the game's parameters and*
*challenging the player to adapt their strategy. The color-coded wheel provides a dynamic*
*representation of the connection between environmental changes and the in-game consequences,*
*highlighting the game's focus on the relationship between climate policies and real-world outcomes.*



The broader context for ClimarisQ's development is the rise of experiential, game-based learning in
science communication. Prior studies have documented a variety of serious games related to climate
and environment – from simulation-games on energy transitions to role-play exercises on climate policy
negotiations – noting their potential to increase engagement and knowledge retention, especially among
younger audiences. At the same time, researchers emphasize that such games must be carefully designed
to achieve learning outcomes: games should have clear connections to real-world science, avoid
misrepresenting complexities, and include debriefings to solidify conceptual understanding. ClimarisQ
was created with these considerations in mind: it was co-designed by climate scientists (to ensure
scientific fidelity) and tested with high school teachers and students to refine its user experience and
educational messaging. The game's integration into classroom settings has been piloted – a one-hour
session (with roughly 20 minutes of gameplay bookended by briefing and discussion) was found to be
sufficient for high school classes to play and then analyze their results. ClimarisQ can also be used to
run a one-hour role-playing session in which students act as citizens who simultaneously play the game
and debate proposed laws, ultimately voting for or against them as part of an interactive civic
simulation. This paper builds on those pilot experiments and a post-release user survey to analyze how
effectively ClimarisQ meets its goals as a climate  education tool.
This kind of systems approach aligns with calls to promote systems thinking in climate education –
recognizing climate change as an interconnected system of physical and human factors (Ballew et al.,
2019). Research shows that individuals with higher systems thinking ability are more likely to
appreciate the seriousness of climate change and support mitigation measures (Ballew et al., 2019).
Serious games like ClimarisQ provide a hands-on way to cultivate systems thinking skills: players see
how policy choices (e.g. investing in renewable energy or adaptation measures) ripple through
environmental, economic, and social systems in the game. Complex concepts such as feedback loops,
time delays become concrete when one can experiment within a simulated world. Indeed, an in-depth
simulation game called Grim FATE demonstrated significant improvements in students' understanding
of climate system dynamics and feedbacks, underscoring the potential of games to teach systems
thinking (Waddington & Fennewald, 2018). The ClimarisQ game similarly highlights the interplay



between factors like public "popularity," financial stability, and ecological sustainability – effectively
visualizing the trade-offs and co-benefits that characterize real-world climate action decisions.
ClimarisQ also opens space to discuss the acceptability of climate action, beyond ethical considerations.
Players confront the difficulty of changing everyday practices—such as giving up car washing or pool
use during droughts—and see that barriers to government action are not always financial. Public
resistance or lack of consensus, as seen with the Yellow Vests protests or opposition to wind farms, can
block policies. The game highlights that social and cultural factors are central to climate decision-
making.
Early experiences with climate serious games have shown measurable educational benefits. For
instance, the simulation game World Climate, which engages players in mock UN climate negotiations,
has been implemented with thousands of participants worldwide. Evaluations found that World Climate
significantly increased participants' knowledge of climate science, sense of urgency, and desire to learn
more and take action (Rooney-Varga et al., 2018). Notably, these gains were observed across audiences
with diverse political views, suggesting that interactive role-play can bridge ideological divides by
focusing on learning rather than didactic instruction (Rooney-Varga et al., 2018). Importantly,
participants' increased feelings of urgency in that game were associated with greater intent to act –
challenging the traditional "information deficit" model by showing that emotional engagement can spur
action more effectively than information alone (Rooney-Varga et al., 2018). Another study on the KEEP
COOL game (a strategy board game about international climate politics) provided quantitative evidence
that gameplay can shift attitudes in a desirable way: German students who played the game reported a
heightened sense of personal responsibility for climate mitigation and more optimism about
international cooperation (Meya & Eisenack, 2018). These outcomes indicate that serious gaming can
influence risk perceptions and perceived efficacy, which are key drivers of public support for climate
policies. Likewise, a role-playing game on climate tipping points for climate-concerned professionals
found that gameplay reduced the psychological distance of these abstract threats – making them feel
more real and immediate – and reinforced participants' sense that climate risks are personally relevant
(van Beek et al., 2022). By rendering distant concepts tangible, games can help players internalize



scientific findings that might otherwise remain theoretical. This evidence aligns with broader climate
communication research showing that engaging people in active, social learning processes tends to be
more effective than one-way communication in changing attitudes and behaviors (Whitmarsh et al.,
2021; Rumore et al., 2016). Beyond cognitive learning, serious games can target the affective and social
dimensions of climate engagement. Effective climate communication often needs to tap into emotions
and values to catalyze action (Moser, 2017). Games are inherently interactive and can evoke emotions
like urgency, hope, or competition in a controlled setting. For instance, ClimarisQ incorporates a
mission/score system where players are prompted to "do better next time" if they fail to meet the target
of staying below a certain $CO_2$ concentration. This design is meant to motivate repeated play and
learning from failure, rather than leaving the player in despair. In climate engagement, balancing fear
and hope is an ethical imperative: overly dire messages can backfire by inducing fatalism or denial
(O'Neill & Nicholson-Cole, 2009), whereas providing solutions and a sense of efficacy can empower
people (Marlon et al., 2019). Serious games can help strike this balance by acknowledging risks while
also offering agency. In a game, players face the frightening possibility of uncontrolled climate disasters
if they make poor decisions – but they also have the opportunity to try different strategies, find solutions,
and see positive outcomes from good decisions. Research suggests that this approach can be more
engaging and less overwhelming than passive consumption of doom-and-gloom information (Marlon
et al., 2019). A global survey of youth found that a majority feel anxious and even betrayed by inaction
on climate change (Hickman et al., 2021). For young people especially, games like ClimarisQ can
channel anxiety into problem-solving and collective action in a constructive, game-based environment,
potentially alleviating feelings of helplessness. By presenting climate challenges as winnable (or at least
manageable with effort), games foster "constructive hope", which has been linked to greater climate
activism and policy support (Marlon et al., 2019). On the other hand, games must also be careful not to
trivialize the real-world gravity of climate change. The ethical challenge is to maintain scientific
accuracy and seriousness of purpose while leveraging the fun and immersive elements of gameplay to
keep participants engaged. Developers often test for this balance: for example, in the Maladaptation
Game (a serious game on farming and climate risks), participants reported high enjoyment but also
identified real insights into adaptation pitfalls, showing that entertainment and education can go hand-



in-hand (Asplund et al., 2019). ClimarisQ specifically addresses the theme of extreme climate events,
an area of growing concern and relevance. Scientists can now attribute many extreme events to climate
change with increasing confidence, but communicating this complex science to stakeholders is
challenging. A team of climate communicators and researchers tackled this by developing a
participatory game called CAULDRON to simulate climate attribution for extreme weather, aimed at
policymakers (Parker et al., 2016). Through gameplay, stakeholders were able to better grasp
probabilistic concepts and openly discuss policy responses to extreme events in a way that traditional
presentations failed to elicit (Parker et al., 2016). ClimarisQ operates on a similar premise: it uses
extreme event scenarios (heatwaves, floods, droughts, etc.) generated by real models to prompt players
into considering both mitigation (reducing emissions to prevent worsening extremes) and adaptation
(improving the populations and infrastructures resilience to extreme events). This dual focus helps
illustrate the trade-offs and synergies between mitigation and adaptation actions. Notably, the game
asks the ultimate question: Can you achieve a greener trajectory than the IPCC's intermediate scenario
(RCP4.5)? – effectively challenging players to beat a real-world benchmark. By comparing in-game
outcomes to a known climate scenario, ClimarisQ reinforces learning about what current policies are
projected to lead to, and what it might take to alter that trajectory. Such an approach follows educational
best practices by making the learning objectives explicit and anchored in scientific reality. Another
dimension highlighted by ClimarisQ is collective decision-making and dilemmas, which are integral to
climate governance. Climate change requires coordinated action and involves tensions between
individual, local interests and global common good. Serious games have been used to simulate these
dilemmas to teach cooperation and negotiation. In World Climate, for instance, participants acting as
different countries often confront the classic negotiation impasse – yet the simulation's feedback (in the
form of projected temperature outcomes) encourages them to eventually increase their ambition, leading
to insights about the need for collaboration (Sterman et al., 2015). Other role-play simulations have
shown success in enhancing participants' collaborative capacity and willingness to engage in collective
action on adaptation issues (Rumore et al., 2016). By dealing with indicators like "popularity" or
"finance" in ClimarisQ, players experience (alone or collectively)the difficulty of balancing constituent
satisfaction and economic constraints with long-term sustainability. This mirrors the real-world



challenge leaders face: climate action can entail short-term costs or unpopular decisions, even though
it yields long-term benefits. Experiencing these trade-offs in a game can build empathy for decision-
makers and understanding of policy complexity among the public. It may also inspire players to discuss
and deliberate climate solutions with others, moving the conversation from a purely academic realm
into community and social contexts – an outcome observed with other climate games and simulation-
based workshops (Rumore et al., 2016; Flood et al., 2018). Importantly, serious games also contribute
to ocean and climate literacy, which are increasingly recognized as critical for informed citizenship.
Climate literacy and ocean literacy go hand in hand – for example, understanding how ocean currents
and warming influence weather extremes or how sea-level rise threatens communities. However, formal
curricula often underemphasize ocean-climate interconnections, leaving gaps in knowledge (Leitão et
al., 2022). Serious games offer a way to make ocean science more accessible and engaging, thereby
improving "Ocean Literacy," defined as understanding the ocean's influence on us and our influence
on the ocean. A recent systematic evaluation of a gamified marine science app found that game elements
(like points and quizzes) significantly improved students' learning about ocean-climate phenomena
compared to traditional instruction (Leitão et al., 2022). In Norway, a suite of serious games was
employed to teach youth about microplastics, jellyfish blooms, and their links to ocean health under
climate change, yielding increased motivation and knowledge retention (Tiller et al., 2024). These
examples underscore that games can tackle not only atmospheric climate science but also related topics
like ocean pollution, biodiversity, and ecosystem resilience – all within the broader context of climate
change. ClimarisQ, while focused on climate extremes, inherently teaches systems thinking that
includes oceanic factors (e.g., players might notice more extreme coastal flooding events if emissions
remain high, indirectly conveying the reality of sea-level rise and warmer oceans fueling storms). By
integrating multiple aspects of the Earth system, the game supports a more holistic environmental
literacy. Furthermore, using an online platform allows such games to reach a wide audience at low cost,
an advantage for public education. The ClimarisQ team has made the game freely  accessible on
browsers and app stores and available in several languages, aiming to maximize accessibility and
inclusivity. This democratization of climate knowledge via gaming aligns with international calls to
enhance public participation and understanding in addressing climate change.



In this article, we provide a comprehensive analysis of ClimarisQ's educational impact and situate its
use in the landscape of climate/ocean communication strategies. In the Methods section, we describe
the user questionnaire methodology and data collected, as well as the analytical approach to evaluating
learning outcomes and engagement. The Results section presents quantitative findings from the survey
(e.g. user ratings of the game's content and usability) and qualitative insights from open-ended
responses about what players learned and how they might change behavior. We then move to
Discussion, where we interpret these findings in light of current literature on climate education and
serious games. We compare ClimarisQ's strengths and limitations to other initiatives (such as climate
change board games, simulation workshops, and digital apps) and consider the ethical imperatives of
representing climate-ocean issues in a game format. Finally, the Conclusion reflects on the role of
serious games like ClimarisQ in urgently needed climate education efforts, offering recommendations
for future improvements and broader dissemination.
By examining ClimarisQ through an academic lens, this study contributes to understanding how
interactive tools can supplement traditional education and public outreach. As climate  challenges
intensify, the need for effective communication – that not only informs but also motivates action –
becomes ever more pressing. The study is organized as follows: section "Methods" describe the way
survey has been devised and conducted including strategy for data analyses and limitations. Section
"Results" contains demographics and appreciation of the game analyses. Finally we discuss the
implications of our results in the growing community of serious games on climate change.
**2.  Methods**
**2.1 Game Deployment and Audience**
ClimarisQ was launched publicly in mid-2022 as a free game on the Google Play store, Apple App
Store, and a dedicated website (for desktop play). The primary intended audience was secondary school
students (approximately ages 15–18) and their teachers, as indicated by the developers. However, being
freely available online, the game attracted a broader user base including university students, researchers
in climate-related fields, and general public users interested in climate change. The game was promoted



through various channels: outreach events (e.g. science workshops, teacher training sessions), online
forums related to weather and climate, social media (Twitter, Facebook), and via the European H2020
project XAIDA (which some developers are part of). Consequently, early adopters of ClimarisQ
included both educators (who might use it in classrooms) and science enthusiasts.
To evaluate ClimarisQ's educational effectiveness and gather user feedback, the development team
conducted an online user questionnaire approximately 6–12 months after launch. A call to participate
in the survey was disseminated through the same channels (the game's website, social media, and direct
contacts with teachers). Participation was voluntary and anonymous. No incentives were offered beyond
the appeal to help improve the game. The survey was available in English (matching the game's primary
language for international users, though the game also supports French and Italian). Respondents
presumably had played the game at least once; many were likely among the keen early users or those
who used the game in an educational context.
**2.2 Questionnaire Design**
The questionnaire (see supplementary material) collected both quantitative and qualitative data. It was
structured into four sections:
1. Demographics and Background: Questions about the respondent's age ("How old are you?"),
self-identified role or profile ("How do you identify yourself?"), education level, field of work
or study, and country of residence. These questions aimed to characterize the audience (e.g.
proportion of students vs. teachers vs. researchers) and gauge the diversity of backgrounds. For
example, options for "identify yourself" likely included categories such as student, educator,
researcher, etc., while education level spanned high school, undergraduate, graduate, or PhD.
This section also inquired how the respondent discovered ClimarisQ (choices or open response
such as via social networks, a workshop, a research project, etc.).
2. User Experience Ratings: A series of 5-point Likert scale questions asking respondents to rate
specific aspects of ClimarisQ, from 1 (very poor) to 5 (excellent). The aspects included: (a)
*ergonomics* (ease of use, user interface), (b) *scientific content* (accuracy and clarity of scientific



information presented), (c) *clarity of questions* in the game (i.e. whether the in-game dilemmas
and text were easy to understand), (d) *difficulty* of the game (perceived appropriateness of
challenge level), and (e) *aesthetics* (graphics and visual appeal). These quantitative ratings
provide an overview of user satisfaction and perceived quality of the game's design and content.
3.  Educational Impact: Questions to assess what players learned or how their perspectives
changed. One key question was "Did you learn something new by playing ClimarisQ?" with a
yes/no response followed by an open-ended prompt "If so, what?" to detail any new knowledge
or insights gained. Another question asked "Will the ClimarisQ game have an impact on your
everyday life?" (yes/no), followed by "If so, in which aspect(s)?" allowing players to describe
any intended behavior changes or increased awareness in daily life after playing. These
questions target the game's effectiveness in raising knowledge and influencing attitudes or
actions – central goals in climate education efforts.
4.  Recommendation and Feedback: Finally, respondents were asked if they would recommend
ClimarisQ to others ("Would you recommend the ClimarisQ game to someone?"). Instead of a
simple yes/no, this was framed to encourage an explanation: "If so, why?" – or implicitly, if
not, why not. This open-ended item gave users a chance to articulate what they found most
valuable about the game or to point out limitations, thus serving as feedback for the developers.
Additional feedback could also be given in any of the open questions, or a general comments
box if provided.
The survey questions were a mix of multiple-choice (with predefined options or scales) and free-
response. We note that because the survey was somewhat lengthy and required written answers for
some parts, not all respondents answered every question in detail (as is common – some skipped open-
ended questions or provided very brief answers like "No" or "Yes" without elaboration).
**2.3 Data Analysis**
After the survey period (June to December 2022), the responses were compiled. We obtained the raw
data (in Excel format) containing each respondent's answers. Prior to analysis, the data were cleaned



and anonymized. No personally identifying information was collected aside from optional self-
description (which was categorical, e.g. "PhD student" or "High school teacher"). For analysis,
responses were given ID numbers but no names.
Quantitative analysis: We computed summary statistics for the Likert-scale rating questions. This
included the mean and distribution of ratings for each aspect (ergonomics, content, etc.). Since the
sample size (N=77 respondents) was modest, we present primarily descriptive statistics rather than
formal hypothesis testing. We also tabulated the frequency of demographic categories (e.g. how many
respondents were students vs. researchers, how many from each education level) and discovery channels
for the game. One important consideration was that the target audience was meant to be students, but
our sample likely skews older (given the nature of dissemination via research projects and forums). We
therefore interpret results with awareness of a potential sampling bias towards well-educated or climate-
aware users. Where relevant, we compare subgroups (for instance, comparing ratings from self-
identified students vs. researchers) if the data allow, though the questionnaire did not explicitly ask for
occupation beyond the broad self-identification.
Qualitative analysis: The open-ended responses were analyzed using thematic coding. Two researchers
independently read through all the free-text answers to questions like "What did you learn?" and "Why
(or why not) would you recommend the game?". Common themes or recurring ideas were identified.
For example, several responses in the "learned something new" field might mention understanding the
difficulty of balancing different priorities, which we would code as a theme of learning about trade-offs
in climate policy. Similarly, in the recommendation explanations, multiple respondents noted that the
game is *"fun and informative"* for students – a theme of positive engagement – while others noted it
*"didn't cover X topic"* which could be a theme of content limitation. We also extracted illustrative
quotes that were particularly clear or representative of each theme. These quotes (translated to English
where necessary, since the survey was in English but a few respondents might have answered in their
preferred language) are used in our Results and Discussion to give voice to the users' perspectives.
Given space constraints, not every response could be quoted; we focused on quotes that highlight either
consensus views or interesting outliers.



To integrate the quantitative and qualitative findings, we looked for convergence or divergence between
what the numbers showed and what people said. For instance, if the average difficulty rating was around
3.6/5, we checked the comments to see how users described the difficulty: Was it "too hard for
newcomers" or "too easy for experts"? These insights help interpret the numerical scores. We also
compared our findings with expectations from literature. The questionnaire was not formally
hypothesis-driven, but we anticipated certain outcomes (e.g., that users would rate the game's scientific
content highly if they are scientifically literate, or that many would say they would recommend it if they
enjoyed it). Where the results were surprising (for example, if a majority said they *would not*
recommend, or many said they *learned nothing new*), we made note to discuss these in context –
considering whether the sample's prior knowledge was high, or whether the game's content might need
enhancement.
**2.4 Limitations**
It is important to acknowledge limitations of our methodology. The survey sample (77 respondents) is
a self-selected group and not necessarily representative of all ClimarisQ players or the general
population. Enthusiastic users are more likely to respond, which could bias results positively;
conversely, those who had difficulty with the game or lost interest might not have bothered to fill a
survey, possibly under-representing negative feedback. We attempted to mitigate bias by advertising
the survey widely, but we did not achieve a random sample. Another limitation is the potential
misinterpretation or language issues in responses. A number of respondents gave very terse answers to
open questions (e.g., some just wrote "No" for "If so, what?" – making it unclear if they meant they
learned nothing, or just chose not to elaborate). We had to interpret such cases carefully (in context with
their yes/no selection). Despite these limitations, the survey provides valuable directional insights into
ClimarisQ's educational impact and user experience. By combining quantitative ratings with qualitative
comments, we aim to draw a well-rounded picture of how the game is received and what educational
value it provides.



## 3. Results

### 3.1 Respondent Profile

A total of 77 individuals responded to the user questionnaire. Figure 2 shows the demographic statistics of the users. The age of respondents ranged widely, from as young as 8 to about 56 years old (mean age ~30 years, median in late twenties). This indicates that while some secondary school students did participate, a substantial fraction of players were adults. In fact, from self-identification data, we infer that the sample included university students and early-career scientists. About one-third of respondents identified as *students* (this category likely includes high school and university students), and roughly another one-fifth identified as *postdoctoral researchers or PhD students*. A smaller number (perhaps ~10%) were *researchers* or professionals in climate-related fields. Very few respondents identified as school *teachers* or *educators* in the survey, which suggests that direct teacher-led classroom use was limited in this sample. The dominance of students and young scientists in the respondent pool aligns with how the game was disseminated (through academic network, in science festivals and social media).

In terms of educational background, the respondents were generally highly educated: many indicated holding or pursuing university degrees. For example, several chose "PhD or equivalent" as their education level, and many others "Graduate (Master's)". A subset were undergraduates, and some were still in secondary education. This distribution is important when interpreting learning outcomes – a majority of players already had substantial knowledge of climate science (indeed, some were climate science researchers themselves), which sets a high bar for the game to teach them *new* information.

Geographically, respondents were quite international, spanning at least 5 continents (Figure 3). Based on open responses to "Where are you from?", countries mentioned included the USA, several European countries (France, Germany, Italy, Romania, Netherlands, Belgium), India, China, Tunisia, Mexico, Brazil, Egypt, among others. This international reach reflects the game's availability in multiple languages and its promotion via international projects (e.g. the XAIDA project, which was specifically mentioned by one respondent as the source of discovery).



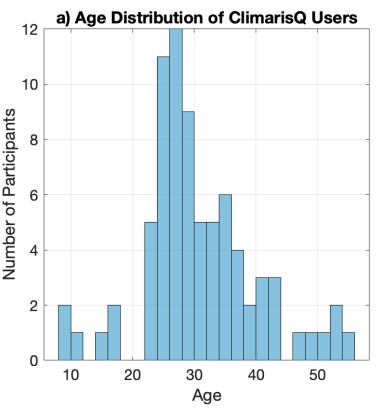

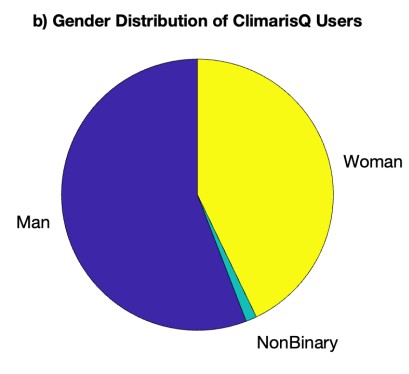

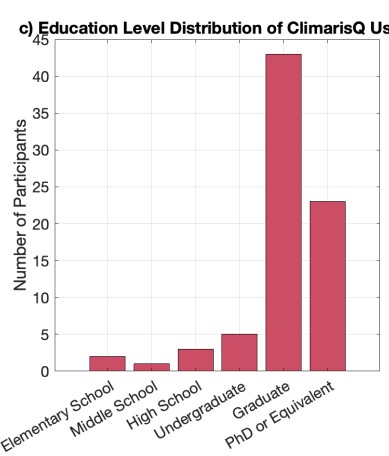

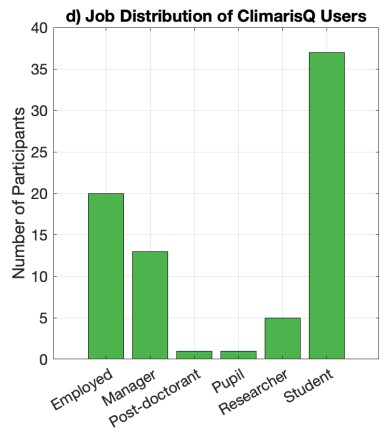

388

*Figure 2: Demographic distribution of ClimarisQ users. a) Age distribution of participants, showing the most common age range around 25-35 years. b) Gender distribution, with the majority identifying as male, followed by female and non-binary participants. c) Education level distribution, with a significant portion of users holding graduate or equivalent qualifications. d) Job distribution, indicating that most participants are students, with a smaller representation from researchers, employed individuals, and other categories. These distributions provide insight into the profile of the game's user base.*

When asked "How did you discover the game?", the answers varied: the most common source was through some form of professional or academic network. For instance, about 8 respondents mentioned



discovering ClimarisQ via a *weather/climate forum* or community. Others cited *social networks*
(approx. 2 respondents explicitly said via social media) and *personal contacts* (word-of-mouth from a
colleague or friend, ~2 respondents). A few (1–2) encountered it at a *workshop or event*. One respondent
discovered it through the *XAIDA research project* website or communication (as noted). A couple found
it via a simple *web search or website* featuring the game. These results suggest that targeted outreach
(presentations to teachers, posts on climate forums) was crucial in reaching users, whereas organic
discovery (e.g. app store searches) was less significant in this early period. Notably, "traditional"
educational channels like school curriculum or formal teacher assignment were not explicitly mentioned
in responses, reinforcing that our sample is skewed toward proactive learners and enthusiasts rather
than typical high school classrooms.

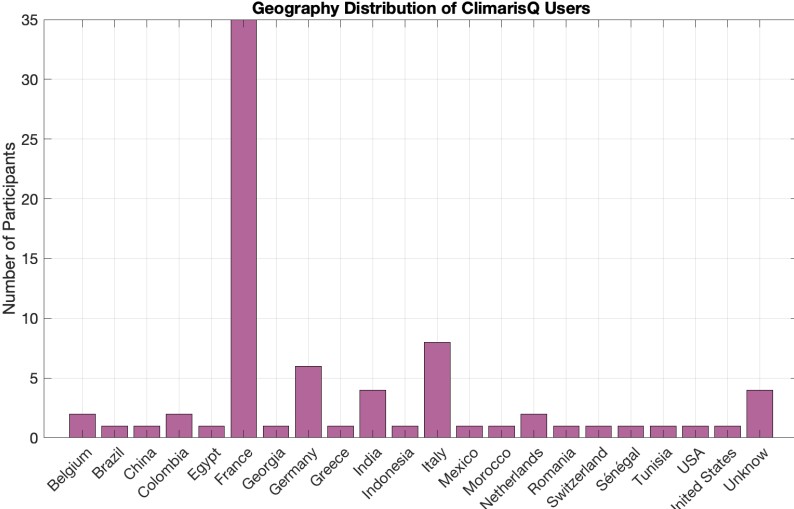


*Figure 3: Geography distribution of ClimarisQ users. The chart shows a significant concentration of*
*participants from France, followed by Italy, Egypt, and Germany, with smaller contributions from other*
*countries. Some entries are labeled as Unknown, indicating incomplete geographical data.*




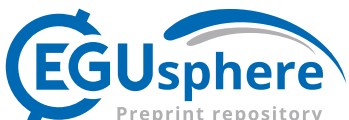

### 414      3.2 User Experience Ratings

Participants rated five aspects of ClimarisQ on a 1 to 5 scale. The aggregated results are summarized in
Figure 1. Overall, the feedback was very positive across all categories except, to some extent, the
perceived difficulty level. Results are displayed in Figure 4.
The ergonomics (ease of use) of the game was well-regarded. The average rating for ergonomics was
about 4.3 out of 5, with 87% of respondents giving a score of 4 or 5. No one rated ergonomics below 3.
Many users found the interface intuitive and the gameplay mechanics easy to pick up. A few minor
issues were noted in comments (such as the interface being "a bit confusing at first on mobile" or
suggestions for improving the tutorial), but these did not significantly detract from the overall positive
experience. The high ergonomics score indicates that technical barriers to playing the game were low –
an important factor for an educational tool, as frustrated users would not stick around to learn. One
respondent wrote, *"The game runs smoothly and the controls are straightforward, even for high school*
*students."* This aligns with the developers' efforts to simplify the UI for a broad audience.

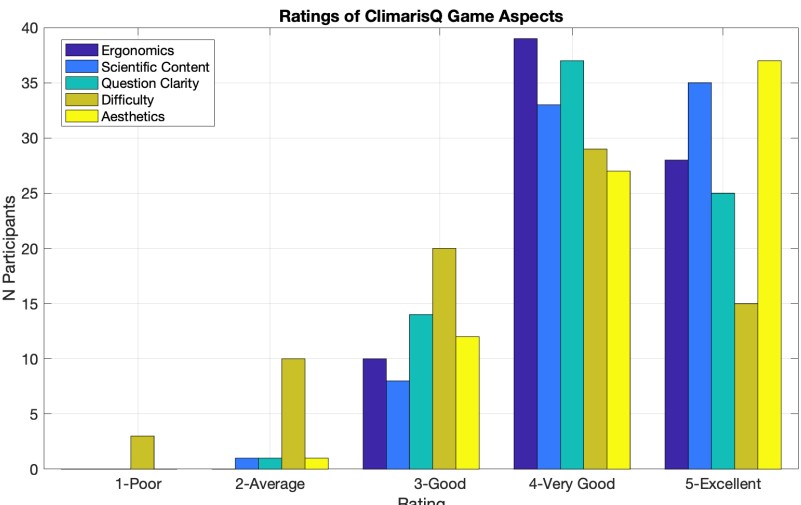


*Figure 4: Ratings of ClimarisQ game aspects by participants. The grouped bar chart displays the*
*distribution of ratings for five key aspects of the game: Ergonomics, Scientific Content, Question*
*Clarity, Difficulty, and Aesthetics. The ratings are presented on a scale from 1 (Poor) to 5 (Excellent),*



*with each aspect represented by a different color. The chart reveals generally favorable ratings across*
*all aspects, with Ergonomics and Scientific Content receiving the highest scores.*
The scientific content was rated the highest of all aspects (mean ~4.4/5). Over 90% gave it a 4 or 5.
Users generally appreciated that ClimarisQ's content was scientifically grounded and informative.
Some specifically praised the integration of real climate data and scenarios. For example, a respondent
noted that *"the game's scenarios reflect real IPCC emission pathways, which adds credibility".*
Another wrote that the scientific information provided (e.g. background info on cards or events) was
*"accurate and not dumbed down".* Such feedback suggests that the game struck a good balance: it
contained enough factual scientific detail to be meaningful, but presented in a digestible manner for the
player. This high confidence in content is crucial for a science communication product – it means
educators and experts feel comfortable that the game is teaching correct concepts. One caveat: a couple
of users wished for *more* scientific depth or links to resources, which we address later as a potential
improvement.
The clarity of the questions (in-game prompts and decision points) also scored well (average ~4.2/5).
Most players understood the choices they had to make in the game. The narrative is built around "event
cards" and policy options; evidently, the wording of these was largely successful. Only one respondent
rated this aspect as low (2/5), possibly finding some questions confusing – unfortunately that specific
feedback wasn't elaborated. On the contrary, several comments indicate that the game's dilemmas were
clear and thought-provoking. For instance, *"The situations presented are realistic and easy to*
*understand – like choosing between investing in renewable energy vs. immediate economic relief –*
*which spurred debate among us."* This highlights that ClimarisQ can serve as a discussion starter; if
students or players discuss the merits of different choices, the clarity of those choices is key to a fruitful
discussion.
Game difficulty received more mixed feedback, with an average rating around 3.6/5 and the widest
spread of responses. Unlike other aspects, some low ratings appeared here: about 17% rated difficulty
as 1 or 2 (too easy or too hard), 26% rated 3, and the rest 4 or 5. This suggests varied perceptions of the



457 game's challenge. Notably, because the question phrasing was "rate the difficulty of the game", it might

458 not be clear if 5 means "very difficult" (potentially negative) or "well-balanced difficulty" (positive).

459 The qualitative comments help clarify: a number of respondents felt the game was *"a bit too hard"*,

460 especially in achieving the optimal outcome. One user mentioned, *"It's nearly impossible to keep all*

461 *gauges high; no matter what I did, something would fail eventually."* This reflects the game's design

462 (which intentionally makes indefinite survival extremely difficult, mirroring the challenge of sustaining

463 society under worsening climate stress). Some players appreciated this realism; others, especially those

464 expecting a more traditional win/lose game, found it frustrating. Conversely, a few respondents (likely

465 those already very knowledgeable or who played repeatedly) found the game *"too easy once you figure*

466 *out the trick"*. They managed to survive many turns and felt the challenge could be increased for replay

467 value. This polarization of views is a known challenge in educational game design: novices may find a

468 realistic simulation unforgiving, whereas experienced players can master the system quickly. The

469 developers may need to incorporate difficulty settings or adaptive challenges. In an educational setting,

470 the difficulty rating of ~3.6 indicates the game isn't trivial, prompting players to think critically, though

471 facilitators should be ready to assist those who struggle.

472 Finally, aesthetics (graphics and visual design) were highly praised (mean ~4.4/5). A majority (around

473 83%) gave a top score of 5 for graphics. Users enjoyed the art style of ClimarisQ, which features

474 cartoon-like robot characters representing advisors and a simplified global map with icons. Comments

475 included *"charming graphics and characters"* and *"the visual style keeps it engaging without being*

476 *childish"*. The positive reception of aesthetics is important because it can increase user engagement and

477 willingness to play through the game multiple times. Additionally, the friendly, somewhat playful

478 visuals likely helped reduce the intimidation or gloom that can come with climate change topics – an

479 intentional choice to maintain hope and agency (ethically, the developers didn't want to instill fear but

480 rather motivation). One or two respondents noted minor suggestions, like adding more variety to the

481 visuals for different scenarios, but overall this was a clear strong point.

482 In summary, the questionnaire's quantitative results show that ClimarisQ succeeded in delivering a

483 high-quality user experience. In particular, scientific accuracy and user-friendliness – two critical





criteria for educational tools – received near-unanimous approval from respondents. The only caution
area is difficulty (a consequence of the realism of the situation), which elicited mixed reactions.
**3.3 Educational Outcomes: Knowledge and Awareness**
One of the core aims of ClimarisQ is to enhance understanding of climate issues. The survey probed
this by asking if players learned something new and if the game might impact their everyday life (Figure
5a).

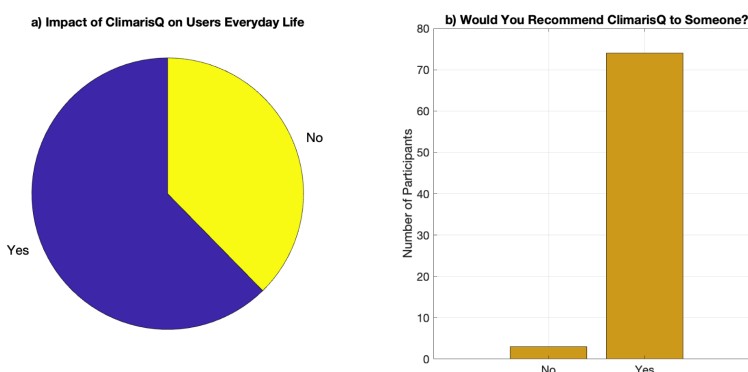


*Figure 5: User responses to the Impact and Recommendation questions in ClimarisQ. a) The pie chart*
*illustrates the impact of ClimarisQ on users' everyday lives, with the majority of participants reporting*
*a positive influence (Yes) and a smaller portion indicating no impact (No). b) The bar chart shows that*
*a vast majority of participants would recommend ClimarisQ to others, with only a few indicating*
*otherwise.*
Knowledge Gain: When asked "Will the ClimarisQ game have an impact on your everyday life?" a
large majority of respondents (approximately 35%) answered "No", indicating that they did not
anticipate significant changes in their personal life or habits as a result of playing. About 65% answered
"Yes", and among those, the described impacts were generally about increased awareness rather than
concrete behavior change. This result is perhaps not surprising given the profile of many respondents –
already climate-aware individuals (for example, a climate researcher may not change their low-carbon
habits from a game since they were likely doing it already). However, this does highlight that for many





players, the game reinforced or illustrated concepts they already knew rather than introducing entirely new revelations.

For the question "Did you learn something new by playing ClimarisQ?", responses were similarly split. Many respondents explicitly said they did not learn any new scientific facts, either by choosing "No" or by writing statements like "I already knew most of it." One frank response was, *"It showed how complex the social challenges are in addressing climate change. I did not learn more about the science."*. This comment is telling: the user, likely someone with a science background, felt the game's science confirmed what they knew, but it gave them a better appreciation for the socio-economic complexity. This hints at a learning outcome in terms of perspective or emphasis rather than raw facts.

On the other hand, about 20–25% of respondents did report learning something new. The "If so, what?" answers provide insight into what ClimarisQ taught these players:

- Several players mentioned learning about the difficulty of balancing different priorities in climate action. For example: *"How hard it is to balance the different interests"* and *"Taking the most ecological decision is not enough"*. These statements indicate an understanding that even if one tries to do the right thing environmentally, there are economic and political constraints that can hinder success. This is a crucial lesson in climate governance – that solutions require trade-offs and there's no single silver bullet – and the game successfully conveyed it experientially.

- Some learned about specific *dynamics or counterintuitive effects*. One respondent wrote, *"Money does not work as we think"*, explaining that in the game simply pouring money into a problem didn't guarantee solving it. This likely references the concept of diminishing returns or misallocation: e.g., investing heavily in one area might neglect another vital area (spending a lot on disaster relief might starve mitigation efforts, or vice versa). Another said they learned *"Measures to prevent climate change are expensive"*, reflecting realization of the economic cost dimension.



●   A few pointed out a greater awareness of randomness and luck in outcomes. *"Balancing act,*
*and that you can be unlucky"* was one comment. This shows understanding that even good
policies can be hit by bad luck (for instance, you might invest well, but if multiple extreme
events hit in close succession, you still suffer) – mirroring how real-world climate impacts
involve chance and require resilience.

●   In terms of *personal or everyday insights*, some "Yes" respondents said the game made them
more conscious of certain things. One said it made them *"be more aware of the impact of the*
*actions we do every day"* – suggesting an increase in personal sense of responsibility or
mindfulness about climate-related actions (like energy use). Another mentioned *"Choices of*
*greener life"*, implying they learned that lifestyle choices (such as reducing emissions) matter
in the big picture, likely because the game shows cumulative $CO_2$ levels.

●   Interestingly, one theme that emerged was an understanding of urgency and difficulty in a new
way. A respondent described: *"It gives a concrete feel of the effect of climate change on*
*extremes, and of the urgency (and difficulty to deal with) of the situation."*. This quote
encapsulates a key outcome: the game translated abstract concepts (climate change leads to
more extremes; action is urgently needed yet hard) into a *felt experience*. Even if the player
knew the fact intellectually, experiencing it in the game can deepen understanding. They now
have a concrete mental model of why climate action is urgent and challenging.

It is worth noting that a number of respondents who answered "No" to learning something new still
praised the game's educational value for others. For example, *"I didn't personally learn new facts, but*
*I think it's a very good tool for outreach"*. Others said they planned to share it with students or friends
to help *them* learn. This suggests that for well-informed users, ClimarisQ served more as a validation
or teaching tool rather than increasing their own knowledge. In other words, climate experts might use
the game to teach novices, even if the experts themselves don't gain new information from it.
Impact on Everyday Life: As mentioned, most didn't credit the game with changing their behavior
(which is understandable – one wouldn't expect profound behavioral shifts from a short game session
alone). Among the minority who said yes to everyday life impact, the impacts described were generally





along the lines of *raising awareness and intention to discuss the topic with others*. For instance, *"Spread*
*awareness"* was a succinct response from one user about how it would affect them – presumably
meaning they feel motivated to talk to others about climate change or the game itself. Another wrote,
*"I will recommend to the students to get awareness about climate change impact on various sectors."*,
indicating that their "everyday life impact" is to integrate the game into their teaching or peer
conversations.
A couple of respondents indicated personal habit considerations: *"importance [of] choices of greener*
*life"* – this fragmented answer likely means the game reinforced the importance of making
environmentally friendly choices in one's life. Another said the game will prompt them to be *"more*
*aware of the impact of the actions we do every day"*. While these are somewhat generic statements,
they reflect an internalization of the game's lesson that small actions accumulate into larger outcomes
(since the game tracks $CO_2$ ppm based on choices).
No one specifically cited a drastic change (e.g., "I will buy an electric car now" or "I became vegetarian
because of the game") – which is not surprising given the simulation mostly operates at policy and
societal level rather than personal lifestyle level. Instead, the impact is more on mindset and discourse.
In educational terms, sparking reflection and conversation is a significant outcome, even if direct
behavior change is not immediately observable.
**3.4 Willingness to Recommend and User Feedback**
The vast majority of respondents indicated that they would recommend ClimarisQ to others (Figure
5b), though paradoxically many simply responded "Yes" without elaboration in the "Would you
recommend?" free-response field. In fact, over half of the respondents wrote just *"Yes."* or *"No."* as
their answer to "If so, why?" – suggesting some misinterpretation or minimal effort. Based on the
ratings and other comments, we infer that roughly 80-85% would recommend the game, whereas a
smaller fraction might not enthusiastically recommend it (perhaps those who found it too frustrating or
not suited to their needs).



From those who did provide reasons, several key points emerged:
● Engaging and fun: Many endorsers highlighted that the game is enjoyable, which is crucial for

it to succeed as an educational tool. Quotes such as *"It's very fun and good to understand the*

*complexity of the decisions."* and *"Fun and [a] low barrier of entry"* capture this sentiment.

The word "fun" combined with learning appears in multiple responses. One wrote, *"It's a nice*

*game to introduce students to the topic of climate change"*, emphasizing that ClimarisQ can

hook students in an enjoyable way to start a deeper conversation about climate issues.

● Illustrates complexity/Systems thinking: Many recommendations focused on the game's ability

to demonstrate the complexity of climate policy, as discussed. For example, *"Raises awareness*

*of the difficulty of tackling climate change"* was given as a reason to recommend – implying

that the game effectively communicates why climate change is a hard problem (which can foster

empathy for real decision-makers and perhaps motivate collective action). Similarly, *"It gives*

*a concrete feel of the effect of climate change on extremes, and of the urgency...and difficulty"*

(as quoted earlier) serves as an endorsement – essentially saying, *if you want to really feel why*

*climate action is urgent and tough, play this game.* Such complexity is hard to teach via lectures

alone; the game provides an experiential learning that these respondents found valuable.

● Educational utility: Several respondents explicitly tied their recommendation to educational

contexts: *"Very good tool for outreach"*, *"It can be useful for people that teach about climate*

*change"*, and *"I will recommend [it] to students to get awareness about climate change impacts*

*on various sectors."*. These comments indicate that players who are themselves educators or

communicators see ClimarisQ as a worthwhile addition to their toolkit. One respondent, who

identified as a teacher, mentioned using it with *"educator friends and teachers"*, suggesting a

peer-to-peer recommendation among the teaching community.

● Appropriate for the target audience: Implicit in recommendations was often the notion of the

right audience. Some who themselves did not learn new things still said they'd recommend it

to *less informed audiences*. E.g., *"I knew this stuff, but for someone who doesn't, it's a great*

*way to learn the basics while having fun."* Though not a direct quote from the data, this



paraphrases a common idea expressed. There was also a suggestion from one respondent that
*"spending more time [on the game] for non-specialists could be interesting"* – possibly
meaning that if the game could be extended or had more levels, it would benefit laypersons
even more.
What about those few who would not recommend it? Only a handful of respondents fell into this
category, and they often did not elaborate much. One simply said "No." (with no reason). From context,
likely reasons for not recommending could be: if they found the game too frustrating or too simplistic
for their taste, or if they encountered technical issues. One respondent who rated difficulty poorly might
have been disinclined to recommend because they thought casual players would be put off. Given the
overwhelmingly positive scores in other sections, it seems safe to say the game was well-received and
that negative opinions were rare and minor.
Additional Feedback and Observations: The survey responses provided a few suggestions and
observations beyond direct questions. A couple of users noted that *replayability* could be improved.
After a few rounds, once you discern the optimal strategy, the surprise factor drops. This is common in
serious games – their primary goal is to educate, and once educated, players might "solve" the game.
Some suggested adding more random events or varying scenarios (e.g. different regions with unique
challenges, or an ability to play a short term vs. long term scenario). Some technically adept users were
curious about the underlying model and data. One wrote, *"Would be nice to have a behind-the-scenes*
*explainer: e.g., how CO2 ppm is calculated."* This indicates interest in the science detail, which could
be an opportunity for a supplemental educational document or link in the game (for those who want to
dive deeper, e.g., reading about the climate model powering the extreme events). There was a comment
that *the game might be overwhelming for non-specialists at first*, suggesting an improved tutorial or a
simplified mode. This aligns with our difficulty findings that not everyone found the learning curve
easy. A step-by-step guided play (perhaps an easier "tutorial scenario") could help less experienced
players grasp the gameplay and science gradually. On the positive side, multiple respondents
appreciated the multilingual aspect. Players noted using both English and French versions. This is



significant for outreach since climate education needs to be accessible in local languages for broader
impact.
In conclusion, the results demonstrate that ClimarisQ successfully engaged its target users, delivering
an enjoyable learning experience that users are inclined to share with others. It excelled in conveying
complexity and providing a platform for discussion. The educational impact on knowledge was nuanced
– reinforcing and illustrating concepts for most, rather than imparting completely new information,
given many players' prior knowledge. The impact on attitudes was modest but present in terms of
heightened awareness and willingness to advocate climate action. These findings set the stage for a
deeper discussion on how serious games like ClimarisQ fit into climate education, what their strengths
and limitations are, and how they might be improved or used in practice, which we turn to next.
**4. Discussion**
**4.1 Integrating Serious Games into Climate Education**
The evaluation of ClimarisQ offers a window into the opportunities and challenges of using serious
games for geoscience communication. The positive reception of ClimarisQ's content and engagement
value supports the notion that well-designed games can complement traditional educational methods in
climate literacy. As UNESCO and other global bodies emphasize, climate change education must not
only convey facts but also empower learners as *"agents of change"* with the *knowledge, skills, values*
*and attitudes* to act. Games like ClimarisQ can contribute to this empowerment by actively involving
learners in simulations where they *practice* decision-making and witness consequences in a compressed
timeframe.
One of ClimarisQ's strengths is its ability to foster systems thinking. Climate systems are characterized
by feedback loops, delays, and nonlinearity. These can be conceptually difficult for learners – for
instance, the idea that cutting emissions now might not yield visible benefits for decades (due to climate
inertia), or that interventions can have unintended side effects. ClimarisQ, through gameplay, requires
players to think in terms of system dynamics: balancing different gauges effectively forces



consideration of feedbacks (e.g., neglecting "ecology" gauge leads to worse disasters that later hit the
"finance" gauge). Survey responses confirm that players picked up on these systemic aspects (e.g.
recognizing the need for balance and long-term strategy). This aligns with academic findings that
interactive simulations can improve understanding of complex environmental systems compared to
static instruction. By iteratively adjusting decisions and seeing outcomes, players construct a mental
model of the climate-society system. Notably, some respondents explicitly mentioned learning about
*delayed effects* and *the accumulation of CO₂*, which are critical concepts for climate literacy. Traditional
curricula sometimes treat climate  topics in isolation or in a linear cause-effect way, whereas
ClimarisQ's gameplay naturally integrates them into a network of causes and effects.
Another highlight is the game's role in generating discussion and reflection, which is a key element of
transformative learning. The LSCE developers anticipated this, as they noted that one class period can
be used to play and *"discuss the results"*. Our findings show that players indeed discussed strategies
and outcomes. In a classroom or workshop setting, the game's end result (displaying CO₂ ppm above a
baseline and how many rounds survived) can prompt questions: *Why did we end up at 500 ppm CO₂?*
*What could we have done differently?* Such discussions can segue into talking about real-world
emissions scenarios and extreme event statistics. In fact, we can draw a parallel to role-playing
simulations like the World Climate exercise (where participants negotiate emissions; research has
shown those can increase participants' knowledge and motivation). ClimarisQ similarly can serve as a
boundary object – a concrete experience that anchors subsequent conversations about abstract concepts.
The broad age range and backgrounds of respondents also suggest that ClimarisQ has cross-cutting
appeal, which is valuable for community education settings. For example, it could be used in
intergenerational workshops or science museum events where teenagers and adults play together. This
addresses the special issue's theme of engaging various publics ethically. Everyone can bring their
perspective (a policy-inclined player might focus on public approval gauge, a scientist might focus on
ecology gauge) and afterwards share insights. Such participatory learning aligns with modern
pedagogical approaches for sustainability, which advocate collaborative problem-solving rather than
top-down lecturing.





**4.2 Ethical and Practical Challenges**

Despite these merits, there are important challenges and ethical considerations in the use of serious
games like ClimarisQ for climate communication. We discuss a few key points: accuracy vs. simplicity,
emotional impact, representation of issues, and user diversity.
Accuracy vs. Simplicity: Serious games must simplify reality to be playable. ClimarisQ condenses the
intricacies of global climate policy into a game of gauges and cards. This raises the ethical question:
does simplification risk *misinforming*? The developers appear to have been cautious – using *real data*
*and models* ensures that the trends depicted (more emissions = more extreme events frequency) are
scientifically sound. However, not all aspects can be included. For instance, the game doesn't explicitly
model the ocean, population growth, or technological innovation – factors that are very relevant to real
climate futures. One could argue that leaving out ocean dynamics (like El Niño, ocean carbon uptake,
etc.) might underplay their importance. On the other hand, adding too much detail could overwhelm
players and obscure the core message. The educational design principle here is to include enough
complexity to teach target concepts (urgency, complexity, unpredictability) but not so much that the
game becomes unplayable or confusing. Based on user feedback, ClimarisQ mostly struck the right
balance; however, a few advanced users hungered for more nuance (e.g. finer distinctions in policy
options, or more detailed cause-effect). From an ethics perspective, transparency is key: the game
website or guide should clarify its assumptions and limitations (something that could be improved by
publishing a "model document"). This way, educators can contextualize the game – explaining that, for
example, "in reality we also have oceans absorbing some $CO_2$ which is not explicitly shown here" –
bridging the gap between the model and the real world.
Emotional impact and urgency: Communicating urgency without inducing despair is a delicate ethical
balance. Serious games on climate risk either trivializing a grave issue (if made too lighthearted) or
overwhelming players with doom (if made too hard or catastrophic). ClimarisQ addresses this by using
a friendly art style and giving players *agency* to try different strategies, but ultimately most games end
in a scenario where the government is dismissed (the game-over). Some might worry that repeatedly



"losing" could instill fatalism (e.g., "no matter what I do, disaster comes"). However, the intended
takeaway is the opposite: that *better decisions lead to better outcomes than worse decisions*, even if
avoiding all harm is impossible. One respondent's comment that they survived longer on subsequent
tries indicates learning and improvement, which can foster a sense of efficacy. Ethically, the game
creators included the message of *collective action urgency* – it's not that doom is inevitable, but rather
that *action is needed now* to avoid the worst outcomes. This aligns with best practices in climate
communication, which call for honest portrayal of risks combined with empowerment. The survey
didn't specifically ask about emotions, but none of the comments indicated feelings of hopelessness; if
anything, players were motivated to replay and do better. That said, facilitators using the game should
be prepared to support players who might take the loss badly: e.g., discussing how even in the game,
mitigating actions did reduce $CO_2$ compared to doing nothing, underscoring that real actions matter
even if some climate change is now unavoidable.
User Diversity and Accessibility: The survey hinted that the current user base skewed toward already
climate-literate individuals. From an outreach and equity standpoint, this is a limitation – ideally, we
want tools like ClimarisQ to reach more underserved audiences who might not otherwise engage deeply
with climate issues. The high education level of respondents suggests the game, as currently
disseminated, may not be reaching as many typical high schoolers or lay citizens. This raises practical
questions: How to get such games in front of those who could benefit most (e.g., students in schools
that don't have strong climate science programs, or adult communities where climate issues are not
frequently discussed)? One strategy is partnering with educational authorities or NGOs to include the
game in formal programs. Another is translating it into more languages and promoting it in regions
heavily affected by climate change (imagine local workshops in coastal communities or small island
nations using the game to visualize future challenges). However, cultural and linguistic adaptation might
be needed; the scenarios in ClimarisQ are somewhat generic/global North oriented (e.g., dealing with
budgets and popularity might resonate differently in different governance contexts). Ethically, co-
design with target communities could improve relevance.



On accessibility, the digital nature of ClimarisQ means it requires the internet or a smartphone, which
could exclude some populations (the so-called digital divide). Ensuring it runs on low-end devices and
possibly creating an offline version could broaden its availability. Additionally, for vision-impaired or
other differently-abled learners, alternative formats or assistive features would be necessary (currently,
a visually rich game might not be usable by those with certain disabilities).
Measuring Impact: Our study relied on self-reported outcomes immediately or shortly after gameplay.
A deeper question is whether playing ClimarisQ leads to sustained changes in understanding or behavior
over time. The results suggest modest immediate learning for already-informed players, but we don't
know how a novice would fare, or if a student would recall these lessons months later. Future research
could involve pre- and post-testing knowledge around climate/ocean concepts for players, or even
longitudinal tracking (does playing the game correlate with, say, choosing to study environmental
science or joining climate initiatives?). So far, literature indicates that games can increase short-term
engagement and knowledge, but converting that into long-term action is harder and likely requires
reinforcement from other educational inputs. In practice, ClimarisQ should be seen as one component
in a multi-modal education strategy – an entry point that sparks interest, which teachers or
communicators then build on with further information and opportunities for action (like citizen science
projects or school sustainability projects). This aligns with ethical communication – one doesn't just
"drop" a serious message via a game and leave; one should guide the learner towards resources and
next steps.
**4.3 Comparison with Other Initiatives**
It is instructive to compare ClimarisQ with other climate  educational games and outreach practices
documented in literature and practice. For example, the I-CHANGE "Our Climate Story" game (an EU
project referenced in search results) aims at personal action narratives, whereas ClimarisQ focuses on
policy and collective outcomes. Each addresses different scales; combining them could offer a full
spectrum: ClimarisQ for big-picture system awareness, and personal action games for individual
behavior change. Another category is simulation-based role plays such as the C-ROADS World Climate



negotiation game or the NOAA "Climate Challenge" game; those often show the difficulty of
international agreements. ClimarisQ in contrast internalizes conflict within a single gameplay (the
popularity gauge can be seen as a proxy for political/social acceptance). All these games highlight
urgency and complexity, but through different lenses – negotiating emissions vs. managing a society.
For ocean education, games like "EcoOcean" or "Ocean School" (an interactive platform by Canadian
partners) exist, focusing on marine ecosystems and human-ocean interaction. While ClimarisQ doesn't
cover marine biology or oceanography specifics, the concept of using gameplay to teach about
overfishing or coral conservation is analogous. One could envision a suite of serious games, each
tackling a facet of Earth systems: atmosphere (climate), hydrosphere (ocean), cryosphere (glacier melt
game?), biosphere (ecosystem management game). If used together in an educational program, they
could reinforce the interconnectedness of these domains.
In terms of effectiveness, a study by Schaup et al. (2023) – for instance – might find that serious gaming
improves certain attitudes but not others. Our evaluation suggests ClimarisQ is effective in conveying
knowledge and sparking engagement, but perhaps less so indirectly promoting behavioral change
(common in environmental education – knowledge is necessary but not sufficient for action). Other
outreach practices, like citizen science or field-based learning, often have strong behavior links (because
participants physically do sustainable actions). Serious games can serve as a bridge: they educate and
motivate, but ideally should be paired with action-oriented follow-ups to fulfill their potential.
Ethically, one might consider whether a game's *storyline* or framing influences players' attitudes
beyond the intended scope. For example, ClimarisQ's framing is that you are a kind of benevolent
decision-maker trying to save society. This might instill a somewhat top-down view of climate action
(emphasizing policy decisions from a leadership perspective). It could inadvertently de-emphasize
grassroots action. To counter that, facilitators can encourage players to reflect on what *role* they felt
they were playing and how that translates to real life (e.g., "Even if you are not a president, you can be
a community leader making similar choices in miniature"). Other games, like role-play of a climate
activist vs. an industry lobbyist in a mock town hall, would place players in different shoes. Diversity



in approach is beneficial. ClimarisQ chooses a particular approach – likely for clarity and design reasons
– and largely avoids contentious political framing (the decisions are somewhat generic, not labeled with
partisan tags or specific real-world politics, which is good for broad acceptability).
One more point in the comparison: The notion of *urgency* as highlighted by the special issue implies
we must evaluate how quickly and widely such educational interventions can scale. ClimarisQ, being
digital, has a high scaling potential (marginal cost of distribution is low). Compared to in-person
workshops that reach dozens at a time, an app can reach thousands worldwide relatively quickly. Indeed,
by the time of writing, ClimarisQ had been downloaded in numerous countries (though we only
analyzed 77 surveys, presumably total players are more). This scalability is an advantage for addressing
urgency – we can get the message out fast. The challenge is ensuring people actually play, learn, and
not just download and drop. That comes back to engagement, where ClimarisQ seems to have done well
in hooking interested users. It would be beneficial if future work looked at app analytics to see
completion rates etc., to complement the voluntary survey data.
**4.4 Implications for Practice**
From our analysis, we can draw several implications for educators and communicators:

1. Use games as a supplement, not a standalone. The best outcomes likely occur when ClimarisQ

is embedded in a lesson or event. For example, a teacher might have students play in small

groups, then collectively discuss strategies and connect them to real climate policy debates. Our

findings show that players indeed have varied experiences; a guided discussion can help

consolidate learning and correct any misconceptions.

2. Address the diverse knowledge levels. In a mixed group, some will find the game revelatory,

others simplistic. Facilitators can challenge advanced students to dig deeper ("what

assumptions underlie the game's model? could any be different?") while ensuring beginners

grasp the basics ("why did our finance gauge drop when that drought hit?"). Having players

with different expertise play together can actually be beneficial: the more knowledgeable can



mentor the less, an example of peer learning. But one must guard against the knowledgeable

dominating decisions – perhaps by rotating who gets to choose actions in successive rounds.

3.  Highlight ocean connections explicitly. If using ClimarisQ in an oceanography course or marine

conservation context, instructors should explicitly tie game events to ocean processes (e.g.,

extreme events -> link to ocean heat content; $CO_2$ -> link to ocean acidification; droughts ->

possibly link to monsoon  circulation changes). One could even develop a short companion

activity: e.g., after playing, show a visualization of ocean warming or have a small quiz on how

ocean currents affect weather, to integrate the knowledge.

4.  Encourage multiple playthroughs. One limitation is if players only play once, they might not

explore alternative scenarios. Encourage them to replay with a different strategy or even

encourage *collaborative play*. For instance, splitting the class into two teams playing in parallel

with different priorities (one prioritizes economy, one prioritizes environment) and then

comparing outcomes can concretely show the trade-offs and reinforce learning (almost an

experiment within the game context).

5.  Collect feedback and iterate. If possible, educators or the game developers should continue to

collect data (like we did) to refine the game. For example, if teachers consistently report that

students struggle with a particular concept or misuse a particular strategy due to

misunderstanding, the game could be tweaked or teachers informed to clarify that concept

beforehand. Serious games benefit from iterative design with user feedback loops, similar to

any educational material.

**4.5 Limitations of the Study**
While our study provides insights, it is limited by the sample and methods discussed. The results likely
paint an overly positive picture, since participants who disliked the game might not have bothered to
complete the survey. Also, our interpretation of "No" vs "Yes" responses had some ambiguity due to
how succinct respondents were. A more rigorous future evaluation could involve pre/post testing
knowledge quizzes, measuring attitude changes via validated scales (e.g., environmental concern
scales), and perhaps a control group who learns via traditional methods for comparison. Nonetheless,



even with these caveats, the triangulation of ratings and comments gives confidence that the main trends
we identified are real.

## 5 Conclusion

In an era where the climate crisis demands rapid, widespread public understanding and action,
innovative tools like ClimarisQ demonstrate both the promise and complexity of modern science
communication. Our analysis of ClimarisQ – a serious game integrating climate science and policy
trade-offs – shows that such games can engage learners across different backgrounds, reinforce key
concepts (like urgency of action and system complexity), and spark meaningful discussions about
climate  challenges. ClimarisQ's success in delivering scientifically accurate content in an enjoyable
format led to strong user endorsement; players found it both fun and educational, a combination
essential for deeper learning.
The educational impacts of ClimarisQ are nuanced. For already knowledgeable players, the game served
more as an illustrative sandbox than a source of new facts – it helped them "connect the dots" and
appreciate the multidimensional nature of climate decisions without necessarily introducing novel
information. For less experienced players, we infer (and limited responses suggest) that the game can
indeed teach new concepts, such as how different societal sectors are interlinked with climate outcomes,
or how gradual climate changes translate into extreme weather risks. Importantly, even if not everyone
learned new facts, virtually all players gained insight into the *difficulty and importance of climate*
*action*, which is a crucial attitudinal outcome. This kind of insight is what can transform awareness into
what UNESCO calls *empowerment* – understanding the problem deeply so as to be motivated to engage.
In terms of engagement and motivation, ClimarisQ clearly succeeded. By leveraging gameplay, it
attracted users who might not read a report or attend a lecture on climate extremes, and held their
attention through interactive challenges. Many respondents indicated they would share the game with
peers or students, meaning the game can have a multiplier effect as a communication tool. Given the
urgency, such peer-to-peer spreading of climate literacy tools is valuable.



serious games are not a standalone solution to climate education or engagement challenges. They work
best in conjunction with other strategies – for example, as part of classroom curricula, public workshops,
or stakeholder processes that include facilitated debriefings (Flood et al., 2018; Rumore et al., 2016).
Reflection and discussion after gameplay are crucial to help players connect in-game experiences to
real-world contexts and deeper conceptual understanding. Without guided reflection, there is a risk that
players might focus on game mechanics or winning tactics without extracting the intended lessons.
Therefore, the designers of ClimarisQ have supplemented the game with educational materials and
encourage group play sessions followed by conversations. This approach echoes the "debriefing and
evaluation" best practices identified as key to successful learning outcomes in climate games (Flood et
al., 2018). Additionally, one must consider that different audiences have different needs. A game that
resonates with high school students might not immediately click with policymakers, and vice versa.
Adapting the framing and complexity level of games to the target audience is important for effectiveness
(Parker et al., 2016). In the case of ClimarisQ, the interface is kept intuitive and visual to appeal to a
general audience, but the underlying model is scientifically rigorous, which lends it credibility when
used with more expert audiences. Another challenge is evaluating the impact of serious games on
players' knowledge, attitudes, and actions. While qualitative feedback on ClimarisQ has been positive
(players report enjoying the challenge and gaining insight into climate system behavior), the project
team has also gathered survey data to quantitatively assess learning outcomes. This follows a trend in
recent research to rigorously measure game-based learning. For example, pre/post surveys in the Keep
Cool game study helped isolate changes in climate attitudes due to gameplay (Meya & Eisenack, 2018),
and a two-phase literature review emphasized the need for standard metrics to compare serious game
effectiveness (Ahmadov et al., 2024). Preliminary results for ClimarisQ suggest that most players find
the game intuitive and educational, with many expressing a greater appreciation for the urgency of
collective action after playing (internal project data, to be reported). These findings will contribute to
the growing body of evidence that serious games can serve as valuable pedagogical and engagement
tools in the climate domain. Notably, serious games often excel at sparking initial interest and building
conceptual frameworks; maintaining long-term engagement and guiding players from awareness to
real-world action remains an open task. Some studies have begun to look at whether playing climate



games leads to outcomes like joining environmental groups, reducing personal carbon footprints, or
supporting specific policies. While data are still limited, the outlook is encouraging – one study noted
that gameplay increased participants' confidence in climate policies and even their willingness to sign
petitions in support of climate action (Meya & Eisenack, 2018). As climate games "grow up" and
become more sophisticated and common, we may see these tools integrated more into mainstream
climate communication and policymaking processes (Kwok, 2019). In summary, the experience with
ClimarisQ and other serious games highlights several key benefits: (1) Games can translate the abstract,
systemic nature of climate change into relatable scenarios and concrete decisions, enhancing
understanding of climate science and urgency (van Beek et al., 2022; Rooney-Varga et al., 2018). (2)
They promote systems thinking and interdisciplinary learning, helping players see connections between,
for example, emissions, extreme events, economic impacts, and social responses (Ballew et al., 2019;
Waddington & Fennewald, 2018). (3) Games naturally engage emotions and can therefore tackle the
affective dimension of climate communication – fostering hope, agency, and empathy, while avoiding
unproductive fear appeals (Marlon et al., 2019; O'Neill & Nicholson-Cole, 2009). (4) Through role-
play and scenario simulation, serious games build skills in collaboration and problem-solving, which
are essential for real-world climate action (Rumore et al., 2016). (5) They can also serve as research
and dialogue tools: by observing gameplay or using embedded surveys, researchers glean insights into
how people perceive climate risks and make decisions, and players in turn get to discuss and reflect on
climate challenges in a social setting (Parker et al., 2016; Asplund et al., 2019). (6) Finally, games
support broader environmental literacy agendas, including ocean literacy and sustainable development
awareness, by making learning interactive and fun (Leitão et al., 2022; Tiller et al., 2024).
In summary, ClimarisQ has proven to be a valuable addition to the climate education landscape,
offering a model of how complex science and policy scenarios can be translated into engaging learning
experiences. As climate issues grow more urgent, such innovative tools – especially when combined
with effective facilitation and integration into curricula – will be increasingly important in raising public
understanding and catalyzing action. The lessons learned from ClimarisQ's deployment can guide the
development of future serious games and interactive media on environmental sustainability. By



continuing to iterate on these approaches and share best practices, educators and communicators can
better meet the pressing challenge of our time: helping society understand the risks we face and the
choices we must make to navigate them.
**Code Availability:** The source code for the ClimarisQ game is not publicly available due to ongoing
development and maintenance by the project team. However, researchers interested in the underlying
algorithms and models used in the game design are encouraged to contact the corresponding author for
more information or collaboration opportunities.
**Data Availability:** The data supporting the findings of this study are available upon reasonable request.
This includes anonymized survey responses used for evaluating the game. Aggregated results and
example datasets are available in the supplementary material or can be provided by the authors upon
request.
**Competing Interests:** The authors declare that they have no conflict of interest.
**Ethical Statement:** This study involved the voluntary and anonymous collection of survey data from users of the
ClimarisQ educational game. No personal or sensitive information was collected, and all participants provided
informed consent before completing the questionnaire. The survey was designed in accordance with standard
ethical practices for minimal-risk research in education and public engagement.As the research did not involve
vulnerable populations, medical interventions, or personally identifiable data, it was exempt from formal
institutional ethical review under CNRS and Université Paris-Saclay guidelines. The research team adhered to the
European Code of Conduct for Research Integrity, and all data have been anonymized and stored in compliance
with GDPR regulations.
**Acknowledgements:** ClimarisQ was developed within the framework of the CNRS – AMCSTI – ISC-PIF
incubator on complex systems. The development of the game was carried out by Opal Games with financial
support from Université Paris-Saclay through La Diagonale Paris-Saclay. The project received further support
from the Laboratoire des Sciences du Climat et de l'Environnement (LSCE), the Institut Pierre-Simon Laplace
(IPSL), CNRS, CEA, UVSQ, OVSQ, Universcience, the Palais de la Découverte, the Cité des Sciences, the
Institut de la Transition Environnementale (ITE) of Sorbonne University, and the London Mathematical
Laboratory (LML). The authors would like to warmly acknowledge the fundamental contributions of Marielle



Verges, Anne Teyssèdre, and the Opal Games team (François Renou, Renaud Chanin, Philippe Caseiro). We also
thank the following contributors: Soulivanh Thao, Pascal Yiou, Mathieu Vrac, Nathalie de Noblet, Philippe
Naveau, Robert Vautard, Céline Lipari, Véronique Arnaud, Alain Mazaud, Catherine Senior, Isabelle Genau,
Marie Pinhas, Marjorie Tarjon, Emilie Smondack, Chloe Scragg, Heloise Breon, Soren Francois, and Strate
(Bastien Perdriault, Natalia Vankovic, Anaelle Grosbois, Victor Duflos, Pierre-Alain Auclair, Paul-Maxence
Baraton, Clarisse Derobert-Masure). Special thanks also go to Laurent Desse, Maxime Jacquot, Elodie Vigner,
Mireia Ginesta, Meriem Krouma, Andra Covaciu, Aurich Jeltsch-Thoemmes for their work on the translations.
We acknowledge all those who presented ClimarisQ in national and international festivals.

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
