# Peer review of "ClimarisQ: What can we learn by playing a serious game for 1"

_EGUsphere, 2025_

## Editor Comment (EC5)

**Preamble**

You have been lucky to receive five reviews (RC) or comments (CC) on your ms "ClimarisQ …".

I will use a single term, review, to refer to all types.

All reviews, overall, concur on most points, especially the important ones.

The following table summarizes the three RCs.

| Sci significance | Sci quality | Pres quality | For final | Look again |
|---|---|---|---|---|
| Good | Fair | Good | Reconsidered after **major** revisions | Willing to review revised ms |
| Good | Good | Good | Subject to **minor** revisions | Willing to review revised ms |
| Fair | Good | Good | Subject to **minor** revisions | Willing to review revised ms |
| Elena S | | | **Major** revision | |

In a word, the 'table says' that the ms is good, but that it needs revision (minor or major).

Your responses (ACs) indicate a real willingness to revise and thus improve your ms.

Thank you for your desire to improve your ms; I wish more authors were like that.

I would like, then, to invite you to revise your ms, and produce v2.

I will then ask the reviewers to take a quick look.

Normally, after that, your ms should be ready for publication.

However, I must warm you that I do not make the final decision.

That is for the Executive Editors, and I have no influence over their final decision. In other words, it will be the quality of your final ms that determines the final outcome.

**Summary of revisions**

Below are some things that I picked out from reading the reviews.

**Terminology**. Most reviewers agreed that being consistent in your use of terminology is important. I think that it is crucial, especially (a) a journal that focusses on 'communication and that prides itself on clarity, and (b) in an article on a topic (games) in which, across the current literature, the use of terms is inconsistent, and – frankly – a mess.

In regard to the use of the word 'game', I personally find it relatively acceptable to use this word as a generic term. My own (idiosyncratic?) preference is to specify somewhere near the start (maybe in a paragraph on terminology) that strictly speaking most climate change (CC) representations are more strictly speaking 'simulations', albeit with game elements. However, most simulation/gaming professionals tend to use the easier term 'game', especially in informal talk. However, as you say in your comments (ACs), your article is not the place to debate the issue.

**Debriefing**. As your simulation/game or game was designed and used with no explicit debriefing, it is difficult to produce pone out of a magic hat just for the article. However, it is important to mention this. Maybe you can insert a paragraph somewhere saying that the next version will include debriefing? Maybe you can suggest, in an appendix or in supplementary materials, some ideas for debriefing?

**Game creators**. Several reviewers mention the need to make explicit that you are game creator, facilitator, data collector and data analyser all together. You mentioned saying in two places. I think that only once is needed – I would say, fairly up front.

**Length**.  More than one reviewer suggests that you should not let length get out of hand.  I do not think that GC research articles have a limit.  However, keep in mind that the longer the article, the less people will read it, at least from start to finish.  I would suggest that you place all 'non-essential' material in an appendix or in supplementary materials.

**What the game looks like and how it works**.  Chloe Lucas raises some important issues, and you have indicated a willingness to clarify many points here.  Other reviewers have also raised such issues.  These issues are important, and – in my view – will help or hinder people's impression of the game, depending on how well you clarify such things.  For this keep your sentences relatively short and relatively simple (split long and complicated ones), use few (complex) subordinate clauses (especially to start a sentence), use active verbs.

**Methodology limitations**.  Several reviewers, especially Elena Shliakhovchuk, highlight what appears to be necessary limitations – necessary because of your overall research design (no questionnaire validation, limited number of respondents).  Such limitations do not invalidate your game, nor your research.  It is just that it should be you who announces them to the reader.  Again, they only need be said once – usually in a section titled 'limitations'.

**Game/education issues**.  André Czauderna's points and issues are worth considering, especially about game design and approach.  However, your article is for a geoscience journal, and cannot step too far out of the geosci realm.

**Impact**.  Pimnutcha P's suggestions regarding impact are really worth including if that is possible.

**CC coverage**.  As you know the area/topic/domain of CC is vast, and no game can possibly focus on, let alone touch on, all topics at once.  I would like to suggest that you delineate your ClimarisQ by including a table or diagram specifying what the game focusses on, what it touches on and what it does not mention.  Examples might be:  ocean acidification, sea level rise, climate justice, denial, various GHGs, ocean currents (eg, AMOC), floods, heat, health problems, illnesses (see articles in the Lancet for examples), shifting climate patterns, drought, starvation, wars, financial aspects, etc, etc, etc.  You might even write a few words for each saying why it is or is not included.

**Other** & **as you revise**.  I have, by design or inadvertently, missed some points, maybe important points.  As you revise you ms, please feel free to reach out to me or to any of the reviewers, with questions or requests for opinions.

**Guides**.  As you revise, please look at these webpages; they will help you.

It is important to follow all the elements in https://www.geoscience-communication.net/submission.html https://oceansclimate.wixsite.com/oceansclimate/writing-guide will help with a number of things, but if you find a contraction with the above GC/Copernicus notes, then those notes are the ones to follow.

I really look forward to this article becoming a great article, one that sets a standard for assessing a climate change game, indeed a geoscience game.